# BENCHMARKING THE ROBUSTNESS OF CROSS-VIEW GEO-LOCALIZATION MODELS

## ABSTRACT

This paper investigates the cross-view geo-localization task, which aims to compare ground query images with an aerial image database tagged with GPS coordinates to determine the capture location of ground images. This task holds considerable significance across multiple domains, including autonomous driving, robotic navigation, and 3D reconstruction. Despite recent notable performance improvements, existing models lack robustness against real-world environmental variations such as adverse weather conditions and sensor noise. This deficiency poses potential risks when integrating this task into safety-critical applications. To comprehensively evaluate the robustness of existing methods, this paper introduces the first benchmarks for evaluating the robustness of cross-view geo-localization models to real-world image corruptions. We applied 16 corruption types to a widely used public dataset, including CVUSA and CVACT, with 5 corruption severities per type, ultimately generating about 1.5 million corrupted images to study the robustness of different models. This study contributes by revealing the performance degradation of cross-view geo-localization models on corrupted images and provides user-friendly robustness evaluation benchmarks. Additionally, we introduce straightforward and effective robustness enhancement techniques (stylization and histogram equalization) to consistently improve the robustness of various models. The codes and benchmarks are available online.

## 1 INTRODUCTION

Cross-view geo-localization is the task of determining the location where a ground-level query image was captured by comparing it with a database of aerial reference images tagged with GPS coordinates, such as satellite images. Traditional image-based geo-localization methods often involve comparing query images with geotagged reference images captured from a ground view. One significant limitation of these approaches is the limited coverage of geotagged images, as they tend to be biased toward well-known tourist destinations. Consequently, ground-to-ground geo-localization methods often fail when reference images are unavailable. In contrast, the reference image dataset for cross-view geo-localization can be created from aerial images captured by devices that densely cover the Earth's surface, such as satellites and drones. As a result, cross-view image matching has become an increasingly popular purely visual geo-localization method. Cross-view geo-localization finds applications in various fields, including autonomous driving (Middelberg et al., 2014), robot navigation (McManus et al., 2014), 3D reconstruction (Häne et al., 2017), and more. Despite its promising applications, the substantial visual differences between ground and aerial perspectives make cross-view geo-localization a highly challenging task.

With the remarkable success of deep learning in numerous computer vision tasks (Krizhevsky et al., 2012; Long et al., 2015; He et al., 2017), recent research efforts (Hu et al., 2018; Liu & Li, 2019; Zhu et al., 2021; 2022) have achieved significant performance improvements on typical benchmarks such as CVUSA (Workman et al., 2015) and CVACT (Liu & Li, 2019). However, existing data-driven deep learning methods often exhibit a sharp decline in performance when confronted with data corruption, such as adverse weather conditions, sensor noise, image blurring, and so on. This issue becomes particularly critical when integrating cross-view geo-localization with safety-critical applications like autonomous driving, where robustness becomes an essential consideration. Figure 1 illustrates the challenges faced by current cross-view geo-localization models when ground-level query images are corrupted. Nevertheless, to the best of our knowledge, current models largely

neglect the evaluation of model robustness, primarily due to the absence of robustness evaluation benchmarks specific to this task within the cross-view geo-localization community.

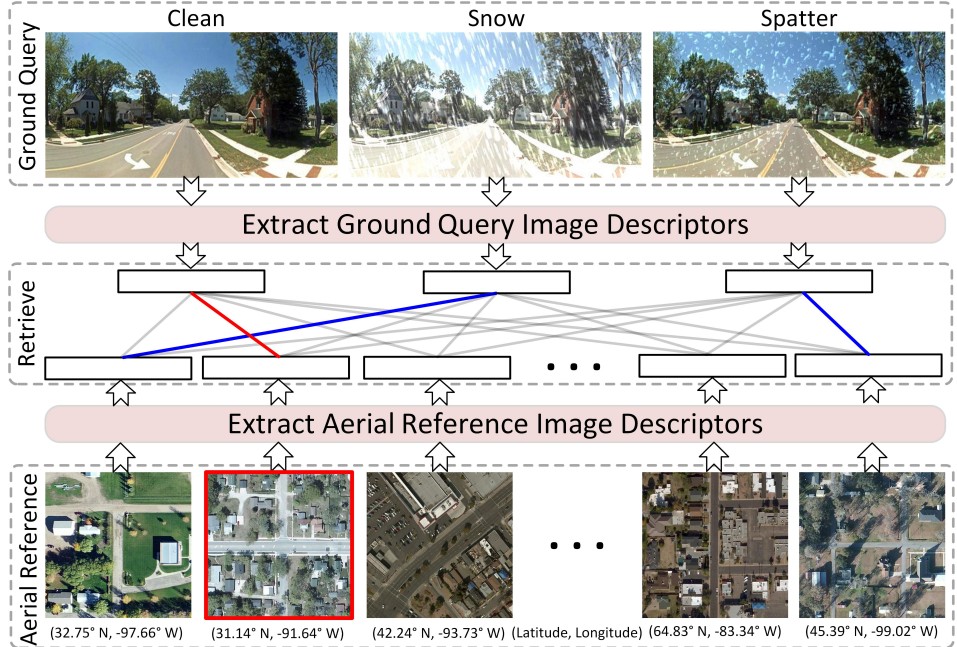

Figure 1: Existing cross-view geo-localization models fail when the ground query image is corrupted. The *red* box indicates the ground truth. The *red* and *blue* lines indicate the retrieved most similar matches. The model that can make the correct match on the clean image fail to match when the ground query image is corrupted.

This paper aims to investigate the performance of existing models when ground query images encounter real-world corruption. To achieve this goal, we propose the first benchmarks for evaluating the robustness of cross-view geo-localization models to real-world image corruption, with the hope of advancing research within the community on the robustness of this task. It is important to note that in this context, robustness refers to the performance of models trained only on clean images when directly tested on corrupted images. We derive our robustness evaluation benchmarks from readily available datasets commonly used for cross-view geo-localization tasks: CVUSA and CVACT (including CVACT_val and CVACT_test), comprising approximately 1.5 million corrupted images.

In this study, we focus exclusively on scenarios involving the corruption of ground images. This choice stems from practical considerations in the deployment of cross-view geo-localization task, where aerial reference images are typically sourced from third-party interfaces and pre-cached by the system. In contrast, ground-level images are captured in real-time from in-vehicle cameras or pedestrian devices. As a result, aerial images exhibit stability and higher quality compared to their ground-level counterparts. Therefore, investigating the robustness of different models when ground-level images are subjected to real-world corruption holds greater practical significance.

In more specific terms, we have categorized image corruptions into 4 major classes, each further divided into 16 subcategories, with each subcategory encompassing 5 severity levels. For both the CVUSA and CVACT_val datasets, we have created distinct evaluation subsets for all corruption types and their corresponding severity level. Consequently, each set corresponds to a total of 80 evaluation subsets, collectively referred to as CVUSA-C and CVACT_val-C. Together, these subsets comprise approximately 1.4 million corrupted images. By evaluating existing methods on CVUSA-C and CVACT_val-C, we gain insight into the fine-grained impact of each corruption type on cross-view geo-localization models, serving as benchmarks for fine-grained robustness evaluation. Simultaneously, we have also introduced comprehensive robustness benchmarks, namely CVUSA-C-ALL, CVACT_val-C-ALL, and CVACT_test-C-ALL. These sets are generated using our

designed model for comprehensive robustness evaluation benchmarks, essentially encompassing all corruption types and severities within a single set, totaling approximately 100,000 corrupted images.

To summarize our contributions in this paper:

- We represent the first comprehensive investigation into the robustness of cross-view geo-localization models. It demonstrates that numerous cross-view geo-localization models experience significant performance degradation when confronted with corrupted images and delves into the robustness of various models.

- We propose the first robustness benchmarks for cross-view geo-localization. These benchmarks are derived from the CVUSA and CVACT (CVACT_val and CVACT_test) datasets. It comprises fine-grained robustness evaluation benchmarks, denoted as CVUSA-C and CVACT_val-C, and comprehensive robustness evaluation benchmarks, denoted as CVUSA-C-ALL, CVACT_val-C-ALL, and CVACT_test-C-ALL. In total, these benchmarks consist of approximately 1.5 million corrupted images.

- We introduce straightforward yet effective robustness enhancement techniques (stylization and histogram equalization) to consistently enhance the robustness of multiple models. At the same time, they without introducing additional training complexity or necessitating architectural alterations to the models.

## 2 RELATED WORK

**Cross-view Geo-localization.** Cross-view geo-localization has witnessed significant advancements in recent years, exploring geo-localization from different perspectives. Initially, Workman et al. (2015) introduced Convolutional Neural Networks (CNNs) to the cross-view matching task, resulting in notable performance improvements. Subsequently, Hu et al. (2018) employed a VGG backbone network with two branches, combined with the NetVlad (Arandjelovic et al., 2016), and proposed the weighted soft-margin triplet loss, achieving state-of-the-art performance. To further enhance network performance, Liu & Li (2019) emphasized the importance of orientation in cross-view geo-localization and devised a method to explicitly provide orientation information to the neural network. Shi et al. (2019) adopted multi-head spatial attention modules to aggregate information-rich and diverse feature maps, while introducing polar transformation for pre-processing to narrow the geometric gap between center-aligned satellite and ground-level images. Recently, Yang et al. (2021) explored the integration of Transformers in cross-view geo-localization, proposing a novel layer-to-layer cross self-attention mechanism that highlighted the significance of considering global dependencies to reduce visual ambiguity. Zhu et al. (2022) introduced an attention-guided pure Transformer approach, which further improved the resolution of satellite images through additional training, thus advancing the performance of task. Zhang et al. (2022) introduced a novel geometric layout extraction module that explicitly decouples geometric information from original features and proposed two types of data augmentation methods. However, all these methods invariably overlooked the performance of models on corrupted images, which is the focus of this paper - robustness.

**Robustness Benchmarks.** Robustness benchmarks are essential in the field of computer vision to enhance the stability and reliability of computer vision systems when facing uncertainties and noise. For instance, IMAGENET-C and IMAGENET-P were proposed to evaluate robustness in classification tasks (Hendrycks & Dietterich, 2019). In the context of autonomous driving, PASCAL-C, COCO-C, and Cityscapes-C were introduced as benchmarks for evaluating the robustness of object detection tasks (Michaelis et al., 2019). Similarly, robustness benchmarks for semantic segmentation were established using PASCAL VOC 2012, Cityscapes, and ADE20K (Kamann & Rother, 2020). Furthermore, KITTI-C, nuScenes-C, and Waymo-C were devised as robustness benchmarks for 3D object detection (Dong et al., 2023). These benchmarks aid in evaluating the robustness of models to accurately comprehend and computer vision under various complex environmental conditions, thus promoting their real-world applications. However, to the best of our knowledge, no researcher has proposed relevant robustness benchmarks specifically for cross-view geo-localization. Therefore, this paper addresses this gap by introducing the benchmarks CVUSA-C, CVACT_val-C, CVUSA-C-ALL, CVACT_val-C-ALL and CVACT_test-C-ALL, thus filling this void in the field.

**Robustness Enhancement.** Several strategies have been proposed to address the impact of corruptions in computer vision. For example, Li et al. (2016) and Fu et al. (2017) present methods

that utilize patch-based priors for background and rain layers, along with a deep CNN to effectively remove rain streaks from images. In a similar vein, He et al. (2010) introduced a simple yet powerful technique employing the dark channel prior to eliminate haze from single-input images. Moreover, Liu et al. (2018) devised the DesnowNet, a multi-stage network specifically designed to remove snow particles from images. However, a limitation of these methods lies in their reliance on specific designs tailored to particular types of corruptions. This hinders their ability to generalize and handle other types of corruptions effectively. As an alternative, some approaches have sought to enhance model performance through data augmentation, incorporating corrupted data during the training process. Although fine-tuning on specific corruption data has shown promise in boosting performance for those particular corruption types Vasiljevic et al. (2016), Geirhos et al. (2018b) discovered that fine-tuning on one type of corruption often struggles to generalize to other corruption types. In another study, Geirhos et al. (2018a) report training on a stylized ImageNet (Deng et al., 2009), which increase overall robustness to different corruptions. In this work, we employ the stylization and histogram equalization to train set, and we observe improvements in robustness.

## 3 METHODOLOGY

### 3.1 IMAGE CORRUPTION MODEL

**Robustness.** First, we consider a set of ground-to-aerial image pairs $\{\mathbf{I}_i^g, \mathbf{I}_i^a\}, i = 1, ..., N$, where $N$ represents the number of image pairs. The superscripts $g$ and $a$ respectively denote ground images and aerial images. For a ground query image indexed as $q$, we assume the existence of ground image encoders $f_g$ and aerial image encoders $f_a$, which have been trained on samples from distribution $\mathcal{D}$. We let $P_{\mathcal{C}}(c)$ approximate the frequency of real-world corruption. Existing models mostly assess performance when samples are drawn from distribution $\mathcal{D}$, denoted as $\mathbf{P}_{(\mathbf{I}^g, \mathbf{I}^a) \sim \mathcal{D}}(d(f_g(\mathbf{I}_q^g), f_a(\mathbf{I}_q^a)) < \{d(f_g(\mathbf{I}_q^g), f_a(\mathbf{I}_i^a)) | \forall i \in \{1, ..., N\}, i \neq q\})$, $d(\cdot, \cdot)$ representing the $L_2$ distance. However, in practical deployments, systems often need to operate on low-quality or corrupted images. Hence, we construct an evaluation of the corruption robustness of model, denoted as $\mathbf{E}_{c \sim \mathcal{C}}[\mathbf{P}_{(\mathbf{I}^g, \mathbf{I}^a) \sim \mathcal{D}}(d(f_g(c(\mathbf{I}_q^g)), f_a(\mathbf{I}_q^a)) < \{d(f_g(c(\mathbf{I}_q^g)), f_a(\mathbf{I}_i^a)) | \forall i \in \{1, ..., N\}, i \neq q\})]$.

**Image Corruption.** Common corruptions are categorized following the approach of Hendrycks & Dietterich (2019), dividing ground-level image corruptions into 4 major and 16 minor categories. The first major category is weather-related corruption, which encompasses snow, frost, fog, glare, and sunny conditions. The second major category consists of blur-related corruptions, including defocus, glass blur, motion blur, and zoom blur. Defocus blur occurs when an image is out of focus, while glass blur arises when images are captured through frosted glass windows of sensors. Motion and zoom blur occurs in scenes with rapid camera movement or quick approaches towards objects, and they are particularly prone to occur in image collections from vehicular devices. The third major category involves noise-related corruptions, which encompasses Gaussian, speckle, impulse, and speckle noise. Gaussian noise emerges under low-light conditions, attributed to the discrete nature of light and causing electronic noise. Impulse noise results from bit errors, resembling the color analog of salt-and-pepper noise. Speckle noise, which occurs due to light interference, leads to the appearance of bright and dark speckles. Lastly, the fourth category includes digital corruption, including contrast, pixelation, and JPEG corruption. Contrast corruption is dependent on lighting conditions and the colors of the objects during capture. Pixelation occurs during upscaling low-resolution images, and JPEG corruption arises when images undergo lossy compression, introducing compression artifacts. We present visualizations of various types and severities of common corruption in Figure 2.

**Fine-Grained Robustness Benchmark Generation Model.** We create fine-grained robustness benchmarks comprising 16 corruption types, each corresponding to 5 severity levels denoted by integers from 1 to 5, where higher numbers indicate more severe corruption. Consequently, a single original ground query image can yield 80 ($16 \times 5$) corrupted images. In the fine-grained robustness evaluation model, each type of corrupted image forms an evaluation subset, with no overlap.

**Comprehensive Robustness Benchmark Generation Model.** Additionally, we introduce comprehensive robustness evaluation models. Prior efforts often generated separate test sets for each corruption type and its corresponding 5 severity levels. While this approach allows for a finer-grained evaluation of each corruption type's impact on the model, it imposes significant storage and computational costs. Recognizing that real-world concerns may prioritize a model's overall performance against diverse corruptions, for each corruption type and severity level, we aggregate all forms of

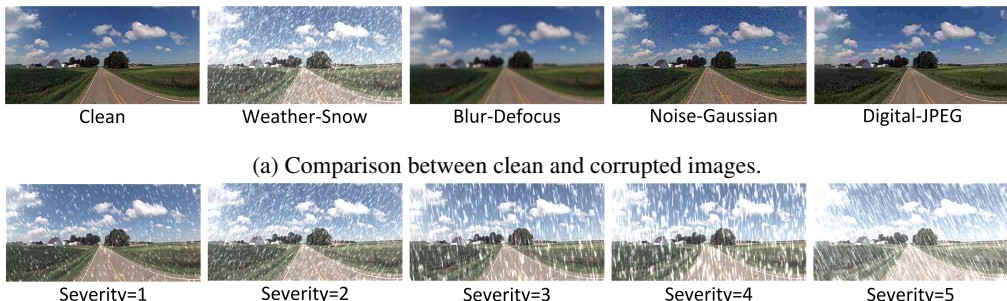

| Clean | Weather-Snow | Blur-Defocus | Noise-Gaussian | Digital-JPEG |

(a) Comparison between clean and corrupted images.

| Severity=1 | Severity=2 | Severity=3 | Severity=4 | Severity=5 |

(b) Comparison of different levels of severity, taking snow as an example.

Figure 2: Various types and severities of common corruption in our robustness benchmark, after cropping are provided (best viewed when zoomed in on the screen). Visualizations of all corruption types can be found in the *appendix*.

corruption into a single evaluation set, creating comprehensive benchmarks. An illustrative diagram of the robustness benchmark generation models is presented in Figure 3.

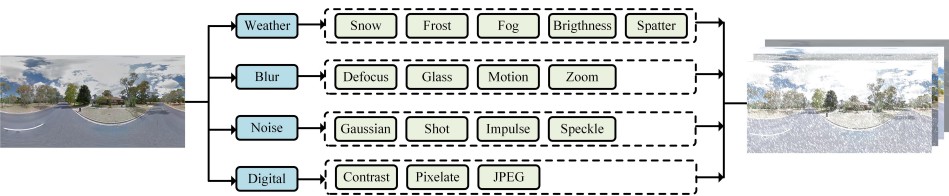

Figure 3: Robustness benchmark generation models. Each corruption category encompasses 5 severity levels. The primary distinction between fine-grained and comprehensive robustness benchmark generation models lies in whether separate evaluation subsets are created for each corruption category and severity level.

## 3.2 ROBUSTNESS ENHANCEMENT

**Stylization.** Style transfer, introduced by (Gatys et al., 2016), merges the content and style of two different images to generate a novel image that retains the content of the original image while adopting the style of the target image. In the context of image classification and object detection tasks, style transfer has demonstrated its effectiveness in enhancing robustness Geirhos et al. (2018a).

**Histogram Equalization.** Histogram equalization is a fundamental technique used for enhancing image contrast. It operates by redistributing pixel intensity values within an image, stretching or compressing the brightness range, thus achieving a more uniform distribution of pixel intensity values across the entire brightness range of the image. In this study, we employ Contrast Limited Adaptive Histogram Equalization (CLAHE) (Pizer et al., 1987) to enhance the robustness of existing methods. Examples of Stylization / CLAHE applied to CVUSA images are shown in Figure 4.

**Training Strategy.** In this paper, we apply Stylization / CLAHE to the cross-view geo-localization dataset, testing two settings: (1) training using the standard (raw) dataset, (2) replacing all training images with Stylization / CLAHE images, thereby eliminating the use of standard images during training, and (3) having Stylization / CLAHE images participate in the training process alongside the standard images with equal probability at each iteration.

## 4 ROBUSTNESS EVALUATION BENCHMARKS

### 4.1 FINE-GRAINED ROBUSTNESS EVALUATION BENCHMARKS

**CVUSA-C.** The CVUSA (Workman et al., 2015) dataset is one of the earliest cross-view geo-localization datasets primarily collected from suburban areas in the United States. It encompasses a

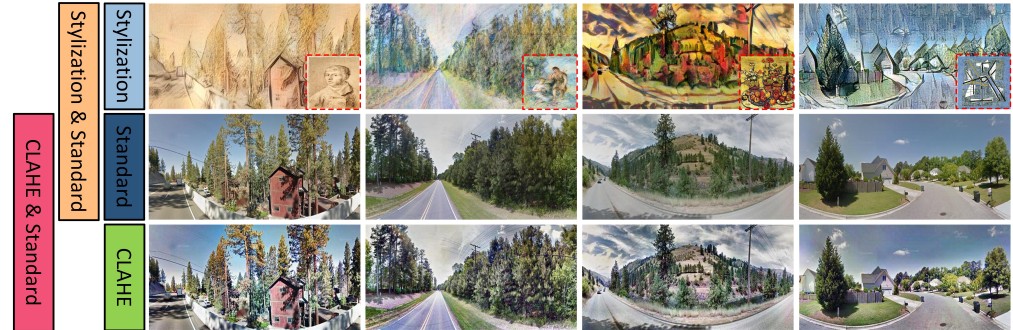

Figure 4: Visualization of Stylization / CLAHE applied to CVUSA dataset. The illustration depicts standard images (middle row), stylized images (top row), and histogram-equalized images (bottom row). The rectangular sections on the left represent different training strategies. The red dashed boxes indicate the style images sourced from `https://www.kaggle.com/c/painter-by-numbers/`.

total of 35,532 pairs of ground-aerial images for training and 8,884 pairs for testing. We have employed CVUSA test set to generate the CVUSA-C fine-grained robustness evaluation dataset. To be specific, CVUSA-C comprises all types and severities of image corruptions, each residing independently within individual subsets of CVUSA-C. Given that there are 16 distinct corruption types and 5 levels of corruption severity, CVUSA-C effectively comprises 80 evaluation subsets. Each of these subsets contains 8,884 ground-level images for testing, resulting in an aggregate of approximately 710,000 images.

**CVACT_val-C.** The CVACT (Liu & Li, 2019) dataset, curated by Liu & Li (2019), primarily encompasses urban areas in Australia. Similar to CVUSA, this dataset comprises 35,532 pairs of ground-aerial images for training and 8,884 image pairs for validation (referred to as CVACT_val). In this study, we have generated fine-grained robustness evaluation dataset for CVACT_val, encompassing all types and severities of image corruptions, resulting in a total of 80 test subsets comprising approximately 710,000 corrupted images, collectively referred to as CVACT_val-C.

**Evaluation Metrics.** The original evaluation metrics for cross-view geo-localization primarily revolved around R@K ($K \in \{1, 5, 10, 1\%\}$), which represents the probability of correctly identifying the matching image within the top $K$ retrieved reference images based on a the query image. Presently, we denote the performance of model on the original validation or test sets as R@$K_{\text{clean}}$. For each corruption type $c$ and each severity level $s$, we employ R@$K_{c,s}$ to gauge the performance of models under corruption conditions. Finally, by averaging across all corruption types and severity levels, we compute the average performance of model R@$K_{\text{cor}}$ under corruption conditions. In this context, $\mathcal{C}$ represents the ensemble of all corruption sets utilized for evaluation. Additionally, we calculate the Relative Corruption Error (RCE) by measuring the percentage of performance degradation. A higher RCE implies a relatively poorer model robustness.

$$\text{R@}K_{\text{cor}} = \frac{1}{|\mathcal{C}|} \sum_{c \in \mathcal{C}} \frac{1}{5} \sum_{s=1}^{5} \text{R@}K_{c,s}, \text{RCE}_{c,s} = \frac{R@K_{\text{clean}} - R@K_{c,s}}{R@K_{\text{clean}}}, \text{RCE} = \frac{R@K_{\text{clean}} - R@K_{\text{cor}}}{R@K_{\text{clean}}} \quad (1)$$

### 4.2 Comprehensive Robustness Evaluation Benchmarks

**CVUSA-C-ALL and CVAC_val-C-ALL.** For the 8,884 pair test set of the CVUSA dataset and the 8,884 pair validation set of the CVACT_val dataset, we designed comprehensive benchmarks for robustness evaluation, as detailed in Section 3.1. These benchmarks are denoted as CVUSA-C-ALL and CVACT_val-C-ALL, respectively. Each of these datasets systematically encompasses a wide range of corruption types and their corresponding severity levels.

**CVACT_test-C-ALL.** The CVACT dataset additionally provides 92,802 pairs of images for testing purposes (referred to as CVACT_test), showcasing dense coverage of urban areas. Similar to CVUSA-C-ALL and CVACT_val-C-ALL, we have generated a comprehensive test subset for

CVACT_test, named CVACT_test-C-ALL, comprising a total of 92,802 corrupted images. These comprehensive robustness evaluation benchmarks serve as valuable tools for evaluating the overall robustness of models under all corruption scenarios.

**Evaluation Metrics.** Since the comprehensive robustness evaluation benchmarks represent independent subsets for testing, the recall accuracy ($R@K_{all}$) can be directly employed for evaluation.

## 5 ROBUSTNESS EVALUATION BENCHMARK EXPERIMENTS

The experimental results of the fine-grained robustness evaluation benchmarks are shown in Section 5.1. In Section 5.2, we report the experimental results demonstrating the influence of stylization on the robustness of model. For different benchmarks, we have chosen representative cross-view geo-localization models to evaluate their robustness. Details regarding the selected methods can be found in Table 1. *Supplementary materials* provides additional experimental results (including comprehensive robustness evaluation benchmark and histogram equalization for robustness enhancement experiment results, etc.).

Table 1: The cross-view geo-localization models adopted for corruption robustness evaluation on CVUSA (CVUSA-C and CVUSA-C-ALL) and CVACT (CVACT_val-C, CVACT_val-C-ALL, and CVACT_test-C-ALL).

| Method | Publication | Backbone | Params (M) | FLOPs (G) | dim | CVUSA | CVACT |
|---|---|---|---|---|---|---|---|
| CVM-Net (Hu et al., 2018) | CVPR'18 | VGG16 | 160.3 | - | 4096 | ✓ | |
| OriCNN (Liu & Li, 2019) | CVPR'19 | VGG16 | 30.7 | - | 1536 | ✓ | ✓ |
| SAFA (Shi et al., 2019) | NeurIPS'19 | VGG16 | 29.5 | 40.2 | 4096 | ✓ | ✓ |
| CVFT (Shi et al., 2020b) | AAAI'20 | VGG16 | 26.8 | - | 4096 | ✓ | ✓ |
| DSM (Shi et al., 2020a) | CVPR'20 | VGG16 | 14.7 | 39.3 | 4096 | ✓ | ✓ |
| L2LTR (Yang et al., 2021) | NeurIPS'21 | HybridViT | 195.9 | 57.1 | 768 | ✓ | ✓ |
| TransGeo (Zhu et al., 2022) | CVPR'22 | DeiT-S/16 | 44.9 | 12.3 | 1000 | ✓ | ✓ |
| GeoDTR (Zhang et al., 2022) | AAAI'23 | ResNet34 | 48.5 | 39.89 | 4096 | ✓ | ✓ |

### 5.1 FINE-GRAINED ROBUSTNESS EVALUATION BENCHMARK EXPERIMENT RESULTS

#### 5.1.1 CVUSA-C EXPERIMENT RESULTS

Table 2 presents the results of 8 cross-view geo-localization methods on the robustness evaluation benchmark CVUSA-C. Overall, there is a strong correlation between robustness and accuracy on clean images, but this correlation is not absolute. Models with higher $R@1_{clean}$, such as L2LTR, TransGeo, and GeoDTR, also achieve higher $R@1_{cor}$, which is understandable as different models show consistent performance degradation on corrupted images. However, it is surprising that L2LTR outperforms TransGeo and GeoDTR in $R@1_{cor}$, even though TransGeo and GeoDTR exhibit higher $R@1_{clean}$. This observation differs from the results obtained in Kamann & Rother (2020) and Dong et al. (2023). The discrepancy might be attributed to its deeper network architecture, more parameters, and FLOPS, making it more robust to input variations. This finding emphasizes the importance of considering the robustness of model independently, especially when its performance on clean data is high, as higher clean performance does not necessarily indicate stronger robustness. In Figure 5a, we plot the RCE of models under various corruption types. Based on our experimental results, we draw the following conclusions.

**Impact of Corruption Types.** Based on the results from Table 2 and Figure 5a, we observed that zoom blur, snow weather, and Gaussian noise corruptions have a significant impact on the performance of various cross-view geo-localization models in the CVUSA-C benchmark, resulting in RCE values exceeding 13% for all models. Conversely, the effects of glass blur, JPEG compression, and pixelation on performance are relatively minor. These findings demonstrate the threats posed by adverse weather conditions, fast motion, and low lighting to cross-view geo-localization models. In contrast, most models exhibited less performance degradation under the influence of glass blur, JPEG compression, and pixelation, possibly due to similar corruptions being present in the training dataset, allowing the models to learn prior knowledge about these types of corruptions.

**Performance of Different Models.** Among all the evaluated models, L2LTR exhibited the most outstanding $R@1_{cor}$ performance. Additionally, we observed a synchronized growth trend between

Table 2: The experimental results of 8 cross-view geo-localization methods on the CVUSA-C. We report the R@1 performance of each method under different corruption (obtained by averaging the 5 corruption severities), as well as the average performance R@1$_{cor}$ under all corruption types.

| Method | Clean | Weather | | | | | Blur | | | | Noise | | | | Digital | | | R@1$_{cor}$ |
|---|---|---|---|---|---|---|---|---|---|---|---|---|---|---|---|---|---|---|
| | | Snow | Frost | Fog | Bright | Spatter | Defocus | Glass | Motion | Zoom | Gaussian | Shot | Impulse | Speckle | Contrast | Pixel | JPEG | |
| CVM-Net | 22.47 | 0.86 | 8.42 | 8.37 | 13.75 | 6.11 | 1.06 | 4.81 | 1.47 | 0.23 | 1.82 | 1.18 | 1.28 | 2.32 | 4.75 | 6.89 | 6.23 | 4.35 |
| OriCNN | 40.79 | 7.36 | 6.51 | 7.57 | 21.69 | 22.01 | 20.46 | 26.10 | 19.60 | 10.32 | 17.24 | 13.95 | 19.40 | 14.09 | 7.94 | 28.51 | 27.27 | 16.88 |
| SAFA | 89.84 | 19.32 | 60.42 | 67.63 | 81.96 | 49.86 | 51.24 | 80.56 | 55.49 | 11.44 | 33.04 | 28.51 | 30.37 | 37.59 | 31.67 | 88.05 | 81.15 | 50.52 |
| CVFT | 61.43 | 8.00 | 30.79 | 47.46 | 47.54 | 27.63 | 24.55 | 44.93 | 34.89 | 8.17 | 21.83 | 19.19 | 20.56 | 26.25 | 38.28 | 57.25 | 47.11 | 31.53 |
| DSM | 91.96 | 24.24 | 64.44 | 84.08 | 82.44 | 57.58 | 62.48 | 84.52 | 66.02 | 25.15 | 49.55 | 46.40 | 48.84 | 60.83 | 72.11 | 90.20 | 85.56 | 62.78 |
| L2LTR | 94.05 | 67.19 | 85.00 | 92.64 | 91.61 | 75.24 | 88.35 | 93.13 | 89.33 | 42.07 | 81.32 | 80.29 | 82.88 | 86.54 | 86.36 | 93.64 | 90.56 | 82.88 |
| TransGeo | 94.08 | 29.39 | 69.50 | 70.89 | 85.01 | 64.26 | 80.97 | 92.16 | 85.96 | 40.97 | 72.95 | 70.27 | 74.32 | 83.99 | 43.01 | 93.74 | 90.13 | 71.72 |
| GeoDTR | 95.43 | 44.20 | 84.95 | 92.80 | 93.55 | 73.14 | 82.64 | 93.29 | 76.80 | 27.19 | 68.40 | 64.45 | 68.53 | 78.28 | 74.80 | 94.45 | 90.20 | 75.48 |

Table 3: The experimental results of 7 cross-view geo-localization methods on the CVACT_val-C. We report the R@1 performance of each method under different corruption (obtained by averaging the 5 corruption severities), as well as the average performance R@1$_{cor}$ under all corruption types.

| Method | Clean | Weather | | | | | Blur | | | | Noise | | | | Digital | | | R@1$_{cor}$ |
|---|---|---|---|---|---|---|---|---|---|---|---|---|---|---|---|---|---|---|
| | | Snow | Frost | Fog | Bright | Spatter | Defocus | Glass | Motion | Zoom | Gaussian | Shot | Impulse | Speckle | Contrast | Pixel | JPEG | |
| OriCNN | 46.96 | 13.94 | 6.13 | 3.78 | 29.45 | 40.54 | 31.71 | 39.99 | 37.58 | 24.89 | 34.24 | 32.27 | 39.01 | 33.28 | 4.56 | 44.38 | 42.56 | 28.65 |
| SAFA | 81.03 | 20.03 | 31.66 | 33.19 | 66.99 | 45.60 | 39.83 | 72.87 | 49.86 | 4.62 | 48.66 | 43.68 | 48.82 | 51.61 | 15.91 | 76.90 | 75.83 | 45.38 |
| CVFT | 61.05 | 15.00 | 22.32 | 42.53 | 47.60 | 37.25 | 31.30 | 53.88 | 36.91 | 4.10 | 35.68 | 30.80 | 36.32 | 36.84 | 31.79 | 58.21 | 57.97 | 36.16 |
| DSM | 82.49 | 31.95 | 51.70 | 70.43 | 69.48 | 52.35 | 57.35 | 80.16 | 67.38 | 15.34 | 58.34 | 53.05 | 58.18 | 63.06 | 52.79 | 81.72 | 80.55 | 58.99 |
| L2LTR | 84.89 | 71.03 | 77.93 | 83.50 | 81.17 | 73.78 | 83.98 | 85.07 | 84.00 | 49.79 | 82.20 | 81.19 | 82.98 | 82.23 | 79.15 | 85.07 | 83.40 | 79.15 |
| TransGeo | 84.95 | 47.65 | 58.51 | 32.91 | 72.67 | 67.13 | 81.43 | 84.83 | 81.80 | 36.34 | 81.96 | 80.86 | 82.84 | 83.01 | 22.18 | 84.92 | 83.74 | 67.68 |
| GeoDTR | 86.21 | 48.24 | 71.74 | 83.26 | 84.60 | 61.39 | 79.11 | 85.51 | 73.44 | 8.26 | 75.44 | 73.99 | 77.06 | 80.23 | 55.48 | 86.01 | 85.19 | 70.56 |

R@1$_{clean}$ and R@1$_{cor}$ for all methods except L2LTR. Notably, CVM-Net demonstrated the weakest corruption robustness, which might be attributed to the corrupted images causing disruptions in the aggregation of local image features into global image descriptors by the NetVlad (Arandjelovic et al., 2016) layer. Consequently, the model lacked a comprehensive understanding of the overall image features, resulting in a significant performance decline.

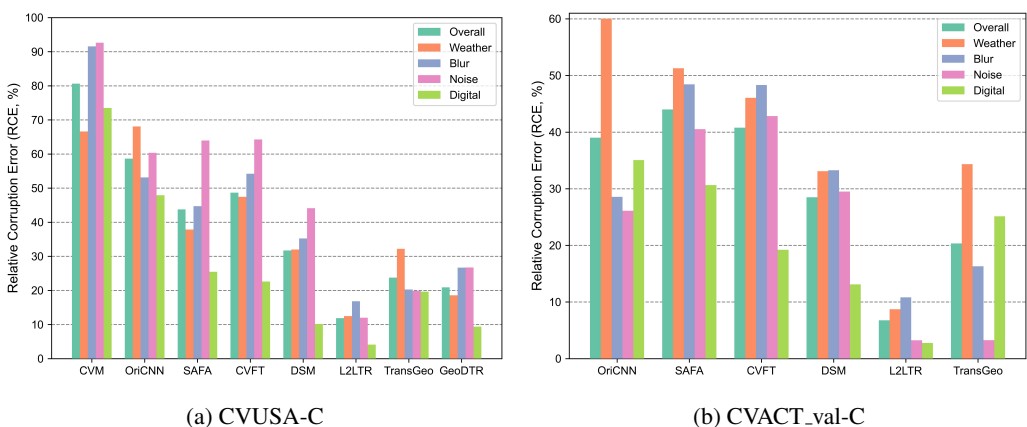

(a) CVUSA-C  (b) CVACT_val-C

Figure 5: The Relative Corruption Error (RCE) of different cross-view geo-localization models on CVUSA-C and CVACT-C datasets. The results are shown for each major category of corruptions, including Weather, Blur, Noise, and Digital, as well as the overall performance across all types of corruptions (Overall).

### 5.1.2 CVACT_VAL-C EXPERIMENT RESULTS

Table 3 presents the results of 7 cross-view geo-localization methods on the robustness evaluation benchmark CVACT_val-C. Similar to the experimental outcomes on CVACT_val-C, except for L2LTR, we observe a strong correlation between corruption robustness and accuracy under clean conditions. Models with higher R@1$_{clean}$, such as L2LTR, TransGeo, and GeoDTR, also achieve higher R@1$_{cor}$, aligning with the consistent trend of performance degradation among different models on corrupted images. Figure 5b shows the RCE of models across various corruption types.

## 5.2 ROBUSTNESS ENHANCEMENT

### 5.2.1 STYLIZATION FOR ROBUSTNESS ENHANCEMENT

We investigated whether the use of stylization enhances the robustness of various cross-view geo-localization models. Following the configuration outlined in Michaelis et al. (2019), we evaluated the performance of 3 classical cross-view geo-localization models on the CVUSA dataset using three different training strategies, as depicted in Figure 4.

Our experimental findings align with those reported by Geirhos et al. (2018b). Training on stylized images indeed resulted in stronger robustness compared to models trained only on the clean images, showing less performance degradation as the severity of corruption increased. However, the performance on the original clean images (severity=0) is noticeably poorer. This can be attributed to the fact that stylized data alters the distribution of the original data. By equally combining stylized and clean data during training, a trade-off between clean and corrupted performance was achieved, leading to both high performance akin to the standard data on clean instances and significantly improved performance on corrupted instances. Results from Figure 6 indicate that both `Standard` and `Stylization & Standard` training enhanced $R@1_{cor}$ under corruption.

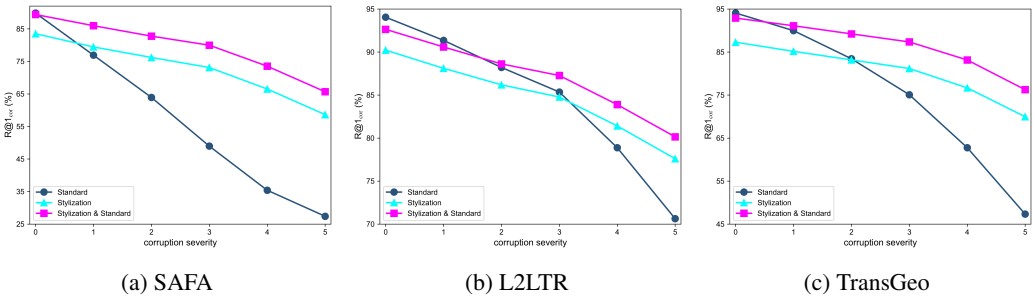

| (a) SAFA | (b) L2LTR | (c) TransGeo |
|:---:|:---:|:---:|

Figure 6: Training on stylized images enhance the robustness of SAFA, L2LTR, and TransGeo on the CVUSA dataset, with each severity level representing the average across all 16 corruption types. Severity = 0 corresponds to clean images for testing . The `Standard` denotes the original, unaltered training data, while `Stylization` denotes training exclusively on images subjected to stylization. `Stylization & Standard` denotes stylized and original training data are equally interleaved during the training process. Notably, the 3 different training strategies require identical training complexity, the experimental configurations and model structures remain consistent throughout.

## 6 CONCLUSION

This paper systematically investigates the impact of ground query image corruption on cross-view geo-localization models. We demonstrate that these models experience significant performance degradation when faced with corruption, a challenge previously overlooked in the context of cross-view geo-localization models. To address this crucial issue and track future developments, we propose fine-grained robustness evaluation benchmarks (CVUSA-C and CVACT_val-C) and comprehensive robustness evaluation benchmarks (CVUSA-C-ALL, CVACT_val-C-ALL, and CVACT_test-C-ALL) for cross-view geo-localization tasks. Extensive experiments are conducted to evaluate existing classical methods on these robustness evaluation benchmarks, revealing the following insights: 1) the corruption robustness of most cross-view geo-localization models is closely related to their clean performance; 2) scaling blur, adverse weather conditions, and contrast degradation significantly impact the performance of various cross-view geo-localization models; 3) glass blur, JPEG compression, and pixelation have a relatively minor effect on performance. Furthermore, we introduce two simple techniques—stylization and histogram equalization to effectively enhance robustness. These techniques without requiring any model adjustments or introducing additional training complexity. We hope that the comprehensive robustness benchmarking, in-depth analysis, and insightful findings presented in this paper will raise awareness within the community regarding the robustness of cross-view geo-localization models and contribute to enhancing their resilience to challenges in complex real-world environments in the future.

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
