# OpenReview forum: "Benchmarking the Robustness of Cross-view Geo-localization Models"
_ICLR.cc/2024/Conference — ICLR 2024 Conference Withdrawn Submission_

### Official Review · Reviewer_dQuR · 2023-10-30

**Soundness:** 2 fair
**Presentation:** 2 fair
**Contribution:** 2 fair
**Rating:** 3
**Confidence:** 4

**Summary:**

This paper presents a benchmark for robust cross-view geo-localization. Necessary experiments have been conducted.

**Strengths:**

This paper presents a benchmark for robust cross-view geo-localization. Necessary experiments have been conducted.

**Weaknesses:**

1. The included perturbations are all simulated. Some real-world perturbations should be considered.


2. The contribution of this benchmark is limited. For benchmark construction, a helpful and convenient tool/page/package should be better for others to use and follow. By doing so, the contribution would be improved a lot.


3. Overall, there is a lack of novelty and contribution to this paper.

**Questions:**

Please see Weaknesses.

---

### Official Review · Reviewer_5XKV · 2023-10-30

**Soundness:** 3 good
**Presentation:** 3 good
**Contribution:** 2 fair
**Rating:** 6
**Confidence:** 5

**Summary:**

1) This paper benchmarks the robustness of cross-view geo-localization models;

2) Specifically, the ground-view images are corrupted under different settings, and the performance of state-of-the-art methods is tested;

3) Two straightforward data augumentation methods (stylization and histogram equalization) are used to improve the robustness of methods;

4) A new robustness benchmark is thankfully received.

**Strengths:**

1) This paper is well-written and easy to follow;

2) Though the technical contribution of this paper is limited, the idea of checking the robustness of existing cross-view model is new and could be useful/important for practical deployment (e.g., autonomous driving);

3) A new robustness benchmark is thankfully received.

**Weaknesses:**

I only have minor comments for this benchmark paper.

1) I want to see some experiments concerning day/night settings. It would be good to provide experiments/examples using low-light ground-view query images;

2) It's good for authors to conduct additional experiments using limited-FOV ground-view query images;

3) Please bolden the best methods in Table 2 and 3;

4) The number/types of style images may impact the performance of data augmentation.

**Questions:**

Please refer to the weaknesses.

---

### Official Review · Reviewer_Lf8Y · 2023-10-31

**Soundness:** 2 fair
**Presentation:** 2 fair
**Contribution:** 2 fair
**Rating:** 3
**Confidence:** 5

**Summary:**

This paper studied the robustness of existing cross-view localization models on real-world image corruptions, including different weather, blur, and noise types, as well as digital corruptions.

This paper augmented the original CVUSA and CVACT datasets with four types of corruptions and evaluated the performance degradation of existing cross-view localization models when trained on clean data. This paper also re-trained the existing models using images with corruption augmentations and assessed their performance on the clean and corruption data, respectively.

**Strengths:**

Studying the robustness of cross-view localization models to real-world corruptions is a practical research direction and is very necessary. This paper augmented the two well-known cross-view localization benchmarks, CVUSA and CVACT, with image corruptions for practical evaluation and analyzed the performance of existing cross-view localization models on these corrupted data.

**Weaknesses:**

While this paper targets real-world problems, which sounds exciting, this paper is more like an experimental report than a research paper. Extensive evaluations of the robustness of previous algorithms to real-world image corruptions are conducted and reported. The types of real-world image corruption are also introduced in an earlier paper (Hendrycks & Dietterich, 2019). Thus, the contribution of this paper is considered insignificant.

This paper indeed investigates an important problem of cross-view localization techniques towards real-world applications. However, the authors should dive deeper and make more insightful contributions rather than simply list the results of previous works.

Furthermore, the evaluation benchmarks are only on CVUSA & CVACT. Zhu et al. (CVPR 2021, VIGOR) have identified the limitations of the two datasets and introduced a more practical dataset, VIGOR. No discussion is conducted on the VIGOR dataset.

Researchers have recently extended the single ground image cross-view retrieval to video retrieval, from the city-scale localization by retrieval to fine-grained localization once the top-1 aerial image has been retrieved. Please find some of these works below. However, no discussions on these topics are presented in this paper.

[1] Shi, Yujiao, et al. "CVLNet: Cross-view Semantic Correspondence Learning for Video-Based Camera Localization." Computer Vision–ACCV 2022: 16th Asian Conference on Computer Vision, Macao, China, December 4–8, 2022, Proceedings, Part I. Cham: Springer Nature Switzerland, 2023.

[2] Shi, Yujiao, et al. "Accurate 3-DoF Camera Geo-Localization via Ground-to-Satellite Image Matching." IEEE Transactions on Pattern Analysis and Machine Intelligence (2022).

[3] Xia, Zimin, et al. "Visual cross-view metric localization with dense uncertainty estimates." Computer Vision–ECCV 2022: 17th European Conference, Tel Aviv, Israel, October 23–27, 2022, Proceedings, Part XXXIX. Cham: Springer Nature Switzerland, 2022.

[4] Xia, Zimin, Olaf Booij, and Julian FP Kooij. "Convolutional Cross-View Pose Estimation." arXiv preprint arXiv:2303.05915 (2023).

[5] Lentsch, Ted de Vries, et al. "SliceMatch: Geometry-guided Aggregation for Cross-View Pose Estimation." CVPR (2023).

[6] Fervers, Florian, et al. "Uncertainty-aware Vision-based Metric Cross-view Geolocalization."  CVPR (2023).

**Questions:**

1. Some terms are confused. For the term "Stylazation", the description in the first paragraph of Sec. 3.2 is consistent with the top row of Fig. 4. However, they differ from the corruption types illustrated in Fig.2 and Fig. 3. Does the training ablation in Sec. 5.2.1 really use the stylization shown in the top row of Fig. 4 instead of the different image corruptions used in the evaluation set?

2. Sec 3.2 and Fig. 4 describes CLAHE. However, there is no experiment on CLAHE.

3. I don't really understand the difference between "Prior efforts often generated separate test sets for each corruption type and its corresponding 5 severity levels (Last para on Page 4)" and the method used in this paper. From my understanding, the experiments in Tab.2 & 3 should be conducted in this way? Why "it poses significant storage and computational costs" compared to the method used in this paper?

4. For the top row of Page 7, it should be "92,802 x 80" rather than "92,802"?